# Bridging Disciplines: Integrating Mental Health and Education to Promote Immigrant Student Wellbeing

**DOI:** 10.3390/bs15091254

**Published:** 2025-09-14

**Authors:** Vanja Pejic, Kristin Russo, Rhode Milord-LeBlanc, Kayla Mehjabin Parr, Sara Whitcomb, Robyn S. Hess

**Affiliations:** 1Department of Psychiatry and Behavioral Sciences, Boston Children’s Hospital, Harvard Medical School, Boston, MA 02115, USA; kayla.parr@childrens.harvard.edu (K.M.P.); sara.whitcomb@childrens.harvard.edu (S.W.); 2Independent Researcher, Boston, MA 02124, USA; 3College of Education and Behavioral Sciences, University of Northern Colorado, Greeley, CO 80639, USA; robyn.hess@unco.edu

**Keywords:** school mental health, immigrant youth, culturally responsive curriculum, interdisciplinary partnership, trauma-informed education

## Abstract

More than 5 million students in U.S. public schools are immigrants or the children of immigrants, highlighting the urgent need for educational practices that honor their lived experiences and promote both emotional and academic growth. This article details a collaborative effort between a school-based psychologist and two high school English teachers to co-design a 12th grade English Language Arts curriculum responsive to the unique strengths and challenges of immigrant youth. Grounded in transformative social and emotional learning, trauma informed principles and culturally sustaining pedagogy, the curriculum weaves together themes of hope, identity, social determinants of health, and agency. The co-development process involved aligning clinical and educational expertise, adapting trauma-informed principles for the classroom, and centering student experience throughout design and implementation. Students reported high satisfaction with the curriculum. Teachers observed stronger student engagement and deeper, more meaningful relationships, attributing these outcomes to the curriculum’s relevance to students’ cultural and community contexts. This case study illustrates the promise of cross-sector partnerships and provides recommendations for creating learning environments where immigrant students can reflect, heal, and thrive through both academic content and emotional connection.

## 1. Introduction

### 1.1. Immigrant and Refugee Children in U.S. Schools

Schools in the United States have a long history of hosting immigrant and refugee children who hail from different parts of the world. Although it is difficult to obtain an exact number, the American Community Survey estimated that in 2021 there were 649,000 children between the ages of 5 and 17 who had been in the United States for three years or less ([19]). There are another 1.5 million immigrant children in this age range who have been living in the United States for four or more years ([44]). It is likely that this number reflects a low estimate since immigrant families are frequently underrepresented in census data. However imperfect, these data suggest that children who are immigrants account for a substantial percentage of students in the United States. Although the largest immigrant populations reside in California, Florida, New York, and Texas, recent immigration patterns have included rural states as well (e.g., Alaska, West Virginia; Montana) suggesting immigrant students can be found throughout the United States ([31]).

Beginning in 2014, the number of immigrant families and children began to rise at an unprecedented rate. Not only did the number of new arrivals increase, but the status of these youth became more complex. Unaccompanied minors from Central America began entering the country, many of them who had experienced violence in their homeland, disrupted education, trauma during their journey, and who were entirely alone in their new country ([18]). In one sample of Central American adolescent refugees, 33% of these youth were experiencing symptoms of PTSD ([6]). Many of the families who are recent immigrants are more likely to come from low-income homes, to have parents with lower levels of education, and whose families may be linguistically isolated ([44]).

The number of new arrivals briefly dipped during the first Trump administration and subsequent pandemic. However, that number shifted upwards again in 2022 with a record number of individuals from Venezuela, Cuba, and Nicaragua. Approximately half of recent immigrant children were reported to be Latino and speak Spanish at home ([44]). A thorough discussion of the immigrant school population is beyond the scope of this paper; however, it is important to note the complexity of the background experiences of immigrant children. The umbrella term ‘immigrant’ may apply to children whose families are researchers and engineers or who are refugees from countries ravaged by war or gang violence.

### 1.2. School as a Critical Context for Adaptation

These students bring many assets to the classroom as they share their language, culture, and lived experiences. The [45] ([45]) has adopted the term ‘newcomer’ to refer to foreign-born students and their families who have recently arrived (sometimes defined as three years or less) to the United States. Although it is recognized that immigrant children have very different backgrounds, the focus of our study was on children who have been forcibly displaced. This broad term refers to individuals who have been forced from their homes for any number of reasons including threats of violence, natural disaster, or even significant climate change. Often displaced children have shared experiences of leaving behind their friends, extended families, culture, and everything that is familiar to come to a new home. The services provided by school personnel can serve as a bridge to help students adjust to their new country and provide social emotional support during this difficult transitional period ([35]).

Consideration of the broad context of newcomer students is critical as their health and wellbeing are reliant on home, school, and community factors. Attention to these factors is especially relevant to immigrant populations as they can often face barriers that can interfere with their successful adaptation to their new countries ([43]). The World Health Organization has referred to these conditions as social determinants of health (SDOH; [47]) with a specific focus on health outcomes. The Center for Disease Control and Prevention (CDC) has adopted SDOH as one of their top three priorities for their Healthy People 2030 initiative ([8]). There are many reasons for adopting this ecological approach to understanding population health with one of the most important being that it is estimated that up to 50% of the variation in health outcomes is related to SDOH ([25]). The five SDOH identified by the CDC include access to and quality of healthcare and education, economic stability, social and community contexts, and neighborhood structure and climate ([8]).

Individualized treatment programming for immigrant youth who have experienced trauma or who are struggling in their new settings is not possible in a school setting, and many newcomer youths do not have access to community-based mental health providers ([40]). Newcomer youth and their families face many barriers to accessing mental health services including cost, lack of health insurance, distrust of institutions, lack of culturally appropriate services, and stigma ([17]). As a result, youth who are from low income and minoritized backgrounds are less likely to access mental health services outside of the school setting ([1]). A focus on systemic variables such as the SDOH related to school and social connection (i.e., access and quality of education, social and community contexts) through mental health promotion and prevention type activities may better serve these youth. Positive youth development programs represent an excellent example of a mental health and wellness promotion approach (e.g., [7]; [16]). Generally, the goals of this type of programming include fostering competency in youth, increasing their self-efficacy, self-determination, and hope, and promoting positive and prosocial behaviors ([7]; [16]).

### 1.3. School-Based Prevention and Promotion

In school contexts, mental health and wellbeing are often defined as students’ ability to manage emotions, build positive relationships, engage meaningfully in learning, and use coping skills to handle everyday challenges. Building on this understanding, school-based promotion and prevention programming are delivered broadly making them accessible to a greater number of students. Mental health promotion programs, which are frequently aligned with positive psychology interventions, focus on supporting individuals’ growth and ability to enhance their own health outcomes ([21]). A recent meta-analysis of health promotion interventions found reduced levels of depression and anxiety, and increased indicators of wellbeing with the largest effect sizes observed in child and youth populations more so than other age groups ([5]). Overall, prevention programming has been shown to be effective for reducing rates of social, behavioral, and academic problems (e.g., [36]). These types of promotion and prevention programming hold promise for reducing negative mental health outcomes and offer an accessible option for diverse groups ([21]; [24]). Many of these types of programs have as their foundation a focus on social emotional competencies.

### 1.4. Project Framework

Central to the work described in this paper are frameworks for prevention programming that are connected to SDOH and comprehensive models of mental health, including Transformative Social–Emotional Learning (T-SEL; [27]). T-SEL builds upon the work of the Collaborative for Academic and Social–Emotional Learning (CASEL), which began in the early 1990s ([12]) and defined competencies for social–emotional learning (SEL) that enable students to thrive in school and in life. These competencies include self-awareness, self-management, social awareness, responsible decision-making, and relationship development. Universal school-based SEL programming is delivered to all students within a class, grade, or school and incorporates systematic instruction and practice of skills associated with the above competencies ([10]; [15]). Some SEL programs are structured with a clear scope and sequence, while others function more as “frameworks” and can be integrated into academic subjects or other courses (e.g., health) which have units that focus on building students’ mental health literacy. This term refers to the knowledge and skills needed to understand, manage, and seek help for mental health concerns, including the ability to recognize signs of distress and know when and how to access support ([30]).

T-SEL expands “traditional” SEL by addressing issues of culture, equity, civics and justice, with an effort to empower commonly underrepresented or minoritized populations and center the voices of youth ([28]). The goal here is to allow every child in a school an equal opportunity to thrive and to refrain from practices that predict differential outcomes based on race, culture, gender, immigrant status, or other demographic variables ([28]). As such, T-SEL programming focuses on the features described in Table 1 ([27], [28]).

To best incorporate a T-SEL framework into programming, researchers suggest anchoring student learning in youth participatory action research (YPAR) and project-based learning (PBL) methods ([27], [28]). These strategies encourage students to pursue relevant questions, critical to helping them understand themselves and the contexts in which they learn and live. There is typically a problem-solving focus to these strategies and iterative processes for reflection and feedback while working to develop a final work sample or product.

T-SEL also is well-aligned with the concept of culturally sustaining pedagogy. Culturally sustaining pedagogy has roots in culturally responsive and culturally sensitive educational practices, which have aimed to recognize and celebrate linguistic and cultural experiences as part of what is a dominant White, Eurocentric approach to education ([33]). T-SEL and culturally sustaining pedagogy expand beyond these roots; however, in valuing and sustaining racial, cultural, linguistic pluralism and supporting students as they learn about within and across group differences as a means to find connection and community ([33]).

Similarly, T-SEL is consistent with principles of trauma-informed education and healing-centered engagement. Any practice that is trauma-informed comes from a space of social awareness from an instructors’ perspective and takes into consideration an individual’s past experiences with violence, natural disasters, displacement, and other traumas in the design of education and programming that meets individual needs and promotes healing ([4]). Trauma-informed practices are based on five general principles outlined by [20] ([20]) and include building safe environments, emphasizing trustworthiness, providing choice, engaging in collaboration, and emphasizing empowerment. Healing-centered engagement extends trauma-informed care in that it empowers systems to recognize the needs of a system, to center wellbeing, and support identity development that is cultural, political, and ecological ([22]).

### 1.5. Interdisciplinary Partnerships

Building partnerships with school staff and preparing non-clinical providers such as teachers to support broad mental health initiatives in school and community settings represent a promising strategy for expanding the capacity to meet student mental health needs ([46]). Teachers or other trained staff can offer many school-based prevention and promotion programs that do not require extensive training for implementation ([23]). In fact, a recent meta-analysis suggests that students may experience more positive outcomes when social–emotional prevention programs are implemented by teachers, rather than clinical staff, such as counselors or psychologists alone ([11]).

One challenge that may arise when relying on teachers to implement prevention programming is the idea that not all teachers are likely comfortable with teaching content related to student social–emotional behavior or mental health. This is unsurprising as teachers report experiencing burnout due to several factors, one of them being student social–emotional behavior ([37]). Further, preservice teaching programs tend not to include specific coursework in instruction related to SEL or trauma-informed care ([41]). Collaboration between teachers and school-based clinical staff may hold promises for implementing meaningful programming that benefits both students and teachers.

When implemented by teachers, school-based interventions that include SEL strategies appear to benefit teachers as well as their students. Engaging teachers in these efforts can enhance their own sense of efficacy and wellbeing, as participation in SEL initiatives has been linked to reduced burnout and greater professional satisfaction ([29]; [38]). In one study, elementary teachers who delivered a comprehensive SEL program reported higher levels of SEL and behavior management efficacy, personal accomplishment, and rated themselves higher on the interpersonal dimension of social–emotional competence as compared to teachers who delivered a behavioral intervention in their classrooms ([14]). To date, most of this research has focused on elementary grade teachers and little is known about the effect of SEL instruction on secondary teachers.

### 1.6. Aim and Purpose

The growing number of immigrant students in public schools who have unmet mental health needs necessitates a novel approach to supporting youth wellbeing. One approach that holds promise is the use of interdisciplinary partnerships to integrate prevention programming such as T-SEL and mental health literacy into academic programming.

The purpose of this qualitative case study was twofold. The first goal was to describe the development and implementation of a curriculum that integrated culturally affirming (transformative) social emotional learning into academic content and made use of culturally sustaining instruction, with a specific focus on the experiences and needs of immigrant and refugee youth. Secondly, the team evaluated the knowledge gains for students and the social acceptability of the curriculum from the perspectives of students who received and teachers who delivered the curriculum. A case study methodology was indicated as researchers were most interested in further defining and exploring how partnership and mental health curriculum integration could occur, a necessary initial step when developing a new way of understanding practice ([39])

## 2. Materials and Methods

### 2.1. Context

The project was based at an alternative high school for newly arrived immigrant and refugee youth in New England. The school serves 525 students from 42 countries including a large number of students from Dominican Republic, Haiti, Cape Verde, Central America (El Salvador, Guatemala, Honduras), and Brazil. The school is organized into three programs: Sheltered English Immersion (SEI) for students who have been in the United States for less than a year; Students with Limited or Interrupted Formal Education (SLIFE); and International. The International program serves a broad range of students who are partially proficient in English Language Development (ELD). The school provides robust programming and support for their students and has a diverse team of educators, mental health providers, family liaisons, and community partners to support diverse academic and psychosocial needs of newcomer students.

### 2.2. Participants

The project curriculum was implemented with three cohorts of 12th grade students enrolled in an English Language Arts course at the alternative school during the 2020–2021, 2021–2022, and 2022–2023 school years. However, it was in the third year (2022–2023) that full implementation of the project occurred. Approximately 225 students participated in the project over the three years of implementation, with 75 participating during the 2022–2023 school year. The student participants were all of immigrant background and reflected the ethnic, cultural, and linguistic identities of their peers across the school community. The students ranged in age from 17 to 21 and were all enrolled in the International program.

### 2.3. Interdisciplinary Project Team

School-Based Psychologist: The school-based psychologist has served as a clinician and a consultant at the school for six years as part of a partnership between the local Children’s Hospital and the school district in which the school is located. The clinician is a former refugee herself and has extensive clinical and research experience in the development and delivery of school-based mental health services for forcibly displaced children and youth. In her current role, she has provided direct clinical care to students and collaborated closely with teachers and staff to implement culturally affirming behavioral health programs and services at the school.

12th Grade English Teacher A: Teacher A has been an educator for 23 years and served as the 12th grade English teacher at the school for 13 years. Teacher A is a white educator and values tailoring her curriculum to the identities and experiences of her students. She has spent her teaching career working in urban, alternative, performance-based schools.

12th Grade English Teacher B: Teacher B has worked in education for 20 years, starting first in adult education teaching English as Second Language (ESL) in community programs. She started at the school as a substitute teacher, before transitioning to a permanent role as a 12th grade English teacher for the last 5 years. Teacher B is a Haitian American and values the importance of bringing her authentic self into the classroom and being someone that her students can see themselves in.

It is important to note that the roles of the school-based psychologist and the English teachers extended beyond their routine responsibilities. In their typical role at the school, the psychologist primarily provides individual and group therapy, crisis intervention, and staff consultation and professional development. However, within the context of this project, they co-developed curriculum content and delivered lessons directly in the classroom, thereby shifting from individualized treatment to proactive, curriculum-integrated mental health promotion. Similarly, while the English teachers’ standard responsibilities involve delivering the grade-level English Language Arts (ELA) curriculum, in this project they systematically embedded mental health literacy and culturally affirming content into reading, writing, and discussion activities. This intentional integration of clinical expertise into academic instruction, and the reinforcement of mental health concepts within ELA coursework, represented a novel interdisciplinary approach that went beyond routine practice.

### 2.4. Exploration: Identifying Needs and Goals

The impetus for this project came from the students themselves. The 12th grade English teachers observed their students’ growing interest in and need for conversations about mental health. This interest was especially apparent during the Senior Capstone Research Project that is part of the 12th grade English program at the school. This assignment incorporates project based learning where students are expected to select a topic of personal interest to research and develop into a comprehensive written paper and an oral presentation. To address students’ interest in mental health topics, the English team partnered with the school-based psychologist to co-develop a revised curriculum that would address some of the key concepts related to mental wellbeing and allow students to explore issues related to mental health and wellbeing in immigrant communities.

In the first year of the partnership, the interdisciplinary team collaborated on one project, the Senior Capstone Research Project. Then, in the subsequent year, the team built on to the Senior Capstone Project by co-developing the year-long curriculum that further enhanced the final unit. In the third year, the introductory classroom lessons offered by the school-based psychologist were incorporated throughout the academic year. Through collaborative planning, the psychologist supported the team in reframing the project’s central inquiry from a focus on distress and mental illness in immigrant communities to a more strengths-based question: “What supports the health and wellbeing of immigrant youth?”

### 2.5. Conceptualization and Framework Alignment

Once the guiding question was established, the school-based psychologist collaborated with the teaching team to identify a guiding framework that would shape the content of the curriculum. Research highlights the critical role of social and environmental context in shaping immigrant youth’s adjustment and mental health outcomes ([43]). To address the central question meaningfully, the team sought a framework that would allow students to explore immigrant mental health and wellness through an ecological lens, emphasizing the dynamic interplay between individual experiences and broader social conditions.

The SDOH framework was selected as the foundation for the curriculum. This approach enabled students to investigate how factors such as housing stability, access to education and healthcare, community safety, and social support contribute to or hinder the wellbeing of immigrant communities ([3]; [47]). By grounding their research in the SDOH framework, students were encouraged to move beyond individual-level explanations and consider how structural and systemic conditions shape mental health and resilience in their lives. By learning about these intersecting systems, students were not only equipped with a deeper understanding of social inequities but also empowered to see themselves as potential agents of change within their own communities.

To ensure cultural relevance, culturally sustaining pedagogy was interwoven throughout the curriculum. This pedagogical approach centers the importance of honoring students’ diverse linguistic, cultural, and historical backgrounds, while also challenging them to engage in critical reflection on systems of power, identity, and representation ([34]). Through this lens, students explored core themes such as acculturation and identity development, resilience, and advocacy drawing meaningful connections between their personal experiences and the broader sociocultural forces that shape immigrant communities.

The curriculum was also aligned with T-SEL, which emphasized developing students’ critical consciousness, cultural identity, and fostering a sense of agency. T-SEL moves beyond teaching discrete SEL skills to help students examine inequities, reflect on their identities, and develop the competencies needed to become change agents in their communities. In this curriculum, T-SEL principles shaped classroom dialog and assignments that asked students not only to understand mental health in context but also to critically analyze how systems of power, privilege, and oppression impact immigrant youth wellbeing. These principles were intentionally integrated into the four core curriculum units that structured the school year: Hope, Identity, Social Conditions, and Agency.

Finally, the curriculum was grounded in trauma-informed educational principles, which acknowledge the pervasive impact of trauma and the essential role schools play in promoting safety, trust, and emotional wellbeing ([32]). Given that many immigrant and refugee students have experienced displacement, family separation, or prolonged uncertainty, this approach emphasized the need to create predictable, supportive learning environments where students could engage fully without being retraumatized ([9]). Trauma-informed practices guided instructional choices such as fostering strong student-teacher relationships, building opportunities for self-reflection, and offering students self-referral slips to meet with the psychologist to attend to diverse emotional needs. These strategies created a classroom climate that cultivated safety, belonging, and resilience.

Together these frameworks, SDOH, culturally sustaining pedagogy, T-SEL, and trauma-informed education formed a cohesive foundation for a curriculum that was academically rigorous and responsive to the students’ cultural, emotional, and developmental needs.

### 2.6. Content Co-Development

With this shared framework, the interdisciplinary team engaged in an intentional process of curriculum design. Across a series of collaborative planning meetings, they worked together to co-develop a curriculum map organized into four thematically organized units: Hope, Identity, Social Conditions, and Agency. The school year began with a unit on Hope, where students examined how hope promotes wellbeing by exploring narratives of individuals and communities sustaining hope in the face of adversity. As seniors preparing for life beyond high school, they discussed personal goals and dreams to set an aspirational tone for the year. Next, the unit of Identity invited the students to critically examine how identities are shaped, affirmed, and marginalized, and to consider how identity development intersects with mental health. The third unit, Social Conditions, shifted focus outward as students began to explore how structural factors such as housing, education, and policy shape opportunities for wellbeing. Classroom dialog encouraged them to identify inequities within these systems and consider how they might be addressed. Finally, in the final unit titled Agency, students were encouraged to use the Senior Capstone Projects to synthesize their learning across the year. Specifically, they were each asked to choose one of the five determinants (i.e., economic stability, education access and language, health and healthcare, neighborhood and environment, family and social connection) to research and identify actionable interventions to support immigrant youth wellbeing and their broader communities. The progression of these themes as outlined in Table 2 was intentional, moving from internal reflection to external analysis, and ultimately to empowered action mirroring the developmental arc of the students themselves as they explored what supports the health and wellbeing of immigrant youth.

Each unit integrated academic content, mental health literacy, and cultural relevance to support cognitive, emotional, and identity development. Students engaged in research, practiced analytical writing, and developed critical thinking skills through texts like Americanah by Chimamanda Ngozi Adichie and other literature by or about immigrants that deepened understanding of migration experiences. Mental health literacy was embedded throughout, beginning each unit with a lesson tied to the central theme and focused on immigrant mental health. Cultural integration ensured students saw their identities reflected in the curriculum. For example, every Friday, students read an “Immigrant Spotlight” featuring individuals making meaningful contributions through social determinants of health (e.g., education, housing, healthcare). This activity reinforced core concepts, modeled resilience and agency, and provided culturally affirming representation.

### 2.7. Implementation

For implementation, the interdisciplinary team adopted a collaborative teaching model. In the early phases, the psychologist met weekly with the teaching team to introduce new concepts, plan lessons, and reflect on implementation. These meetings ensured that mental health content was integrated meaningfully into the academic curriculum while remaining responsive to students’ needs. At the beginning of the school year and the start of each unit, the psychologist facilitated a classroom lesson introducing the unit’s theme and its connection to health, mental health, and wellbeing. These sessions provided students with an opportunity to explore mental health concepts and vocabulary through both the introduction of new content and shared personal experiences. Grounded in culturally affirming pedagogy, the lessons encouraged students to reflect on how identity, culture, and context shape understandings of mental health. Teachers actively participated in these sessions and reinforced key ideas through follow-up discussions, writing assignments, and literary analysis, ensuring a cohesive and integrated learning experience. This approach enabled teachers to build deeper understanding of mental health and wellness promotion as well as enhance their course content by integrating these concepts into the traditional curriculum. To support student wellbeing, the psychologist offered check-in slips during each co-taught lesson. These slips allowed students to request one-on-one time with the psychologist, providing a low-barrier way to access support. This trauma-informed strategy, paired with the culturally affirming content, helped create a supportive classroom environment that centered students’ voices and empowered them to engage both emotionally and academically.

### 2.8. Evaluation and Continuous Improvement

For this case study, only the 2022–2023 year of data were reported as this was the first year that the full program was delivered. However, the curriculum was evaluated through multiple methods during the course of the academic year and at the end. Throughout the year, the interdisciplinary team met regularly to review student questions and academic work, which created an iterative feedback loop that allowed the curriculum to be adapted and strengthened in real time.

### 2.9. Pre/Post Survey

Students completed pre- and post-surveys as part of the Senior Capstone unit assessing changes in their attitudes toward mental health, help-seeking behaviors, and overall mental health knowledge (See Appendix A for Pre/Post Surveys). The pre–post survey was designed to assess proximal outcomes aligned with the intervention content, specifically students’ mental health knowledge and their attitudes toward help-seeking and community health. They were not intended to capture the full scope of student mental health and wellbeing, which is a broader construct encompassing psychological, emotional, social, and functional domains. The survey included 10 statements formatted as true/false knowledge items (e.g., Health includes our physical health, mental health, and social health), 5 Likert scale items designed to assess attitudes (e.g., It is important to learn about health.), and 3 open-ended questions (e.g., Please list some things that people your age can do to improve their health and mental health?). Together, this survey was developed to capture both quantitative and qualitative shifts in understanding and perspective. Student responses to the knowledge items were reported as percentage scores out of 100. All attitude items were measured on a 4-point Likert scale (1 = strongly disagree to 4 = strongly agree), with higher scores indicating greater endorsement with a maximum combined mean score of 4. Given the small sample size and attrition between pre- and post-surveys, inferential statistical analyses were not conducted. The surveys were designed primarily as part of a program evaluation, and results are presented descriptively to highlight observed trends in student knowledge and attitudes.

At the end of the school year, students completed a Satisfaction Questionnaire to evaluate the quality, relevance, and cultural responsiveness of the curriculum. These responses were reviewed by the interdisciplinary team which included the school-based psychologist, the educators, and members of the research team. Rather than conducting a formal qualitative coding process, the team used a structured review approach: responses were read multiple times to identify recurring themes and illustrative statements. Attention was given to capturing both consensus perspectives (ideas that appeared across many students’ responses) and unique insights that shed light on individual experiences. Representative quotations were selected to highlight the range of student voices for statements that reflected students’ perspectives on their learning and the enhanced curriculum. These reflections were then used to contextualize quantitative survey findings and to illustrate how students perceived the curriculum’s relevance, cultural responsiveness, and impact on their wellbeing and learning. In this way, qualitative data were not treated as systematically coded research findings, but rather as student-centered feedback that directly informed considerations for ongoing program improvement.

### 2.10. Teacher Interviews

Semi-structured interviews were conducted with the two English teachers who co-developed and implemented the curriculum. The interviews explored their perspectives on curriculum design, implementation, and the impact on both students and their teaching practices. Interview transcripts and notes were reviewed by the research team. Using a structured review approach rather than formal qualitative coding, the team read responses iteratively to identify preliminary themes and areas of convergence. Special attention was paid to reflections that spoke to the integration of mental health and identity into academic content, the collaborative co-development process, and shifts in teacher practice. Representative quotations were selected to illustrate teachers’ perspectives and to highlight how the curriculum shaped both pedagogy and professional growth.

## 3. Results

### Curriculum Outcomes

Student Pre–Post Survey Outcomes. Over three school years, data have been collected to evaluate and inform the development of the curriculum. In the third year of implementation, all parts of the program were in place, and pre- and post-Surveys were given to assess student knowledge and attitudes towards mental health.

Overall, students showed gains in their mental health knowledge with 90% or more of students answering correctly. As shown in Table 3, some of the most notable changes occurred in student responses to questions about the importance of culture (e.g., “Maintaining my culture helps me stay healthy” increased from 64% to 100%) and perceptions of immigrants (e.g., “Immigrants are healthier, more educated, and commit less crime” increased from 38% to 75%). Other items that began with already high levels of correct responses, such as understanding that “health includes physical, mental, and social health” (95% to 100%) and that “mental health influences how we think, feel, and act” (95% to 100%), remained consistently strong across pre- and post-assessments.

Additionally, students’ attitudes towards the curriculum topics showed small positive gains. For example, ratings improved on “knowledge of strategies to help self” (M = 2.8 to M = 3.2) and “belief in making a community difference” (M = 2.7 to M = 3.1). Together, these results suggest that students not only improved their factual knowledge about mental health but also reported greater confidence in coping, help-seeking, and contributing positively to their communities.

Student Satisfaction Outcomes: The surveys included open-ended questions that prompted students to give their ideas about the negative and positive actions one might take related to mental health (Pre/Post Survey) and encouraged them to share their thoughts on the different lessons offered as part of the curriculum (Satisfaction Questionnaire). Students’ open-ended survey responses and reflections on the curriculum revealed several main takeaways that illustrate how they understood, internalized, and applied the material.

Expanding Understanding of Mental Health: One student reflected, “This lesson taught me about mental health and I never knew about it. I thought only the crazy people had mental health but now I understand everyone has it.” Lessons helped students understand how biology, psychology, and the social environment particularly for immigrant communities influence mental health. As one student shared, “Learning about social environments that impact our mental health helps us become agents of change in our own life and in our community.”

Affirming Cultural Identity and Bicultural Pride: The curriculum emphasized culturally relevant content and affirmed students’ identities. Weekly “Immigrant Spotlight” stories highlighted diverse narratives of resilience and contribution, allowing students to see their own experiences reflected in the material. This approach encouraged pride in their cultural backgrounds and bicultural identities. One student explained, “It’s more good for us immigrants to be bicultural than assimilation, so now I know why it’s more healthy and why it’s important for our health to stay like that.” Another simply stated, “I learned that we should always be proud of who we are.”

Growth in Coping and Help-Seeking: Students also reported more positive attitudes toward mental health and help-seeking. By the end of the unit, more students felt confident identifying coping strategies, seeking support, and helping others. One student captured this growth by saying, “How I can keep myself healthy in life once I am out of my circle where I feel belonging and supported, and how I can support those around me when they experience difficult times. How I can advocate and get in touch with those that are right now working towards helping in this social issue.” Overall, students rated the lessons highly and expressed a desire for more classes like it.

Observations: Informally, it was noted that several students self-referred for individual counseling with the psychologist after participating in this program. Although this might be an indication that more students were struggling, it is likely that they learned about the psychologist’s role as an available support and had gained a better understanding of mental health that helped them to be able to take this step. Collectively, these outcomes suggest that the ELA curriculum created a learning environment where students could explore complex topics, reflect on their identities, and build skills to promote wellbeing for themselves and others.

Teacher Outcomes: Teacher interviews provided valuable insights into the collaborative process, the integration of mental health and identity into academic content, and the ways the curriculum shaped both student engagement and teacher practice. Several central takeaways are outlined below.

Value of Holistic and Relevant Framework: Teacher semi-structured interviews underscored the transformative impact of co-developing and implementing a mental health-integrated, culturally affirming curriculum for immigrant youth. Both teachers entered the project with a shared motivation to better support their students not just academically, but emotionally and socially. They were particularly drawn to the incorporation of the Social Determinants of Health (SDOH) framework, which provided a concrete and holistic lens through which students could examine the relationship between their health, identity, and environment. One of the teachers noted, “I loved working with material and resources that could have a real and direct impact on immigrant youth. The content was authentic, applicable, and critical.” For the educators, this framework offered a way to make abstract concepts like mental health tangible and personally relevant to their students’ lived experiences.

Collaboration and Co-Development are Essential: The co-development process was described as “essential, collaborative and intellectually enriching” for both of the teachers. Teachers highlighted the importance of working alongside a school-based psychologist, whose interdisciplinary expertise in psychology, public health, and trauma-informed care expanded the scope and depth of the curriculum. One teacher noted that the process of cross-examining ideas across disciplines such as anthropology, sociology, psychology, and education was critical to the curriculum’s evolution. The other teacher explained, “I always feel like I need a person to learn from, it was a nice collaboration between teachers and the mental health provider. I would not have been able to envision this project without working with someone like you (the school-based psychologist). It’s not in a binder… it’s unique.”

Centering Cultural Identity and Representation: Culturally affirming practices were central to the curriculum’s design and implementation. Lessons intentionally centered the experiences of immigrant students, with activities like the Immigrant Spotlight and guest speakers who shared their stories in their own languages. As one teacher described, “I brought my family, myself really, immigrants who reached their potential and told the stories in their own language.” Another recalled the power of a guest speaker: “Here is a Haitian police officer… who had to learn English and talked about how protective factors helped her reach her goals. It was very moving.”

Observed Student Engagement: Teachers observed strong student engagement across emotional, social, and academic domains. Students responded positively to the themes presented in the curriculum, particularly those focused on hope, acculturation, resilience, and identity. These topics resonated deeply, often giving students language for experiences they had previously not been able to articulate. One teacher remembered a student saying, “I learned things that are real in your class,” while another recalled a quieter student who reflected, “That immigrants can do stuff… I didn’t know immigrants come to this country and do things.” These moments underscored how elements of the curriculum served as a bridge between classroom learning and students’ lived experiences.

Shifts in Teacher Practice and Pedagogy: The experience also had a profound effect on the teachers themselves. It reshaped their planning, shifting from deficit-based models to asset-based frameworks that center student strengths. One teacher noted, “It gave me more confidence in my own pedagogy… encouraging them, but also realizing that I needed encouragement myself.” Another reflected on the long-term influence, “I’ve always worked with marginalized students, but I learned how our immigrant students are different and deserve a curriculum tailored to who they are.”

Need for Sustained Collaboration: Both teachers emphasized that meaningful integration of mental health and identity into academic content is not only possible, but necessary. However, they cautioned that this work cannot fall on classroom teachers alone. Sustained collaboration with mental health professionals was seen as critical to both the success and sustainability of the project. As one summarized, “You don’t need to be a therapist to do this work. With the right materials, you can still teach all the reading and writing standards while truly supporting students’ wellbeing.

## 4. Discussion

Immigrant youth are a rapidly growing population in U.S. K-12 schools ([44]). Given the psychosocial and acculturative stressors associated with migration, newcomer youth enter school with a range of previous experiences, strengths, and needs. Schools play a vital role in supporting newcomer youth to adjust to living in a new community, particularly by providing mental health prevention and promotion programming ([35]).

Our findings build on and extend existing research in several key ways. First, while prior studies have highlighted the promise of school-based SEL for immigrant youth, these efforts have been limited in the incorporation of culturally responsive pedagogy and are rarely co-developed with immigrant-serving educators ([42]). Our results demonstrate how an SEL-informed curriculum that integrates T-SEL, culturally sustaining pedagogy, and trauma-informed practices can support student identity development and strengthen their mental health literacy.

Second, the majority of existing interventions for this population are led either by teachers or mental health providers, with few emphasizing collaborative, interdisciplinary delivery ([42]). Our project addresses this gap by illustrating how school-based mental health providers can leverage their expertise to build educator capacity, expand the reach of services, and reduce silos through interdisciplinary collaboration. These findings align with recent calls in the literature for integrated, team-based approaches ([26]) and provide a practical example of how such models can be operationalized.

Finally, while translating theory into practice is often challenging, this intervention shows that principles from teaching pedagogy and SEL can be effectively embedded into a year-long curriculum. The length of this enhanced curriculum (i.e., academic year) offered sustained opportunities for students to see their experiences reflected in the curriculum through SDOH framing and culturally relevant practices. This not only echoes literature emphasizing the importance of addressing acculturative stress ([26]) but also adds to it by demonstrating how culturally sustaining and identity-affirming content can foster both immediate classroom engagement and longer-term agency in immigrant youth.

Given the positive outcomes of teacher-delivered social–emotional prevention programs, yet recognizing the undue burden these initiatives can place on teachers, this project illustrated how psychologists can use their role as a consultant and direct service provider to build teacher capacity. Through classroom lessons, the psychologist enhanced both teachers’ and students’ mental health knowledge and provided a shared language for discussing mental health and wellbeing. Consequently, teachers developed a clearer understanding of their role in engaging students in meaningful conversations about mental health and wellbeing as non-mental health providers. Although teachers implemented the majority of the programming, the psychologist’s direct lessons with students allowed them to build trust and rapport in a familiar setting. This increased visibility helped reduce the stigma surrounding seeking mental health support, leading to more student self-referrals. This is particularly meaningful given the barriers to accessing and utilizing care among this population ([13]). Ultimately, this interdisciplinary partnership fostered a culture of mental health awareness and help-seeking within the school community. This type of intentional collaboration represents a sustainable way to reach a greater number of students in a time where schools are plagued by shortages of mental health providers and lack of available services to meet student needs ([2]).

### 4.1. Lessons for Interdisciplinary Collaboration

Establish trust and collaborative partnership. Prior to this project, the interdisciplinary team had not previously worked together. Establishing trusting relationships across different disciplines takes time and effort to develop. To create the context for collaboration, the psychologist intentionally established a presence in the school through observing classes, facilitating roundtable discussions with staff to understand the current needs and existing support for students, and delivering professional development to enhance their understanding of the school culture and to build relationships with the faculty. By integrating into the school community, the psychologist enhanced awareness of their expertise and the breadth of their role, thereby inviting collaboration from staff. Furthermore, because the team focused on one project initially, they were able to establish a collaborative relationship and learn from initial implementation, before expanding the curriculum and the partnership.

Leverage expertise of team members. A key success of the partnership was leveraging the strengths and expertise of both the psychologist and the teachers. The psychologist, with specific expertise in immigrant mental health, guided the framing of the strengths-based essential question, consulted teachers on how to incorporate the SDOH framework, and delivered classroom lessons related to mental health literacy. Teachers utilized their expertise to develop lesson plans and design assignments in accordance with students’ academic levels, English language needs, and core curricular standards. They also brought distinct individual strengths, one teacher drew on her own lived experience as an immigrant in shaping the curriculum, while the other contributed deep knowledge of project based learning and curriculum development. Through their shared work, each individual brought their unique knowledge and expertise to the project. Teachers retained ownership of curriculum design, yet experienced increased confidence, self-efficacy, and capacity to integrate T-SEL, culturally sustaining pedagogy, and trauma-informed educational principles.

### 4.2. Recommendations for Schools and Providers

The success of the enhanced curriculum and interdisciplinary partnership offers valuable insights for providers seeking to implement similar interventions. The curriculum content was developed in direct response to needs voiced by students and teachers, highlighting the importance of centering community voices and experiences. Providers are encouraged to engage in structured conversations, such as needs assessments with students and teachers to better understand areas of interest and needs that can inform intervention content ([42]). Given the critical role of establishing trusting partnerships, it is recommended that interdisciplinary teams begin with a small pilot project. This approach allows teams to develop working relationships and evaluate the acceptability and effectiveness of the program before scaling to a larger initiative. The year-long curriculum enabled students to engage with the material deeply and within a meaningful context, as each unit intentionally built upon the previous one. Thus, it is recommended that trauma-informed, culturally responsive, and T-SEL aligned strategies be embedded as foundational elements of curriculum design to maximize student engagement. To implement this effectively, it is essential that school leadership allocate protected time for interdisciplinary collaboration, including time for planning meetings and intervention co-implementation. It is also recommended that providers and teachers receive professional development to build their capacity to design interdisciplinary interventions, especially for student populations with whom they have limited experience.

### 4.3. Limitations

While the project has notable strengths, there are several limitations. The project was implemented in a unique context where all students identify as immigrants, and the school is equipped with educators, staff and resources to serve this population. Relatedly, all interdisciplinary team members had prior experience working with immigrant students, which may have positively influenced development, implementation, and outcomes. Additionally, pre–post data was only reported for one year and the satisfaction questionnaire is specific to the final unit (Agency). As a pilot project in a unique context, external validity is limited. Future research must quantitatively compare student outcomes between those who receive the curriculum and those who do not enhance the current findings. However, qualitative feedback from both teachers and students suggests the curriculum was effective and impactful.

## 5. Conclusions

This project demonstrates the value of interdisciplinary collaboration in designing and implementing a culturally affirming, trauma informed curriculum that integrates TSEL into academic instruction. By centering student voices, drawing on the SDOH framework, and leveraging the combined expertise of teachers and a school-based psychologist, the initiative created a learning environment that fostered deep learning, reflection and belonging for immigrant youth. Students gained mental health literacy, pride in their identities, and a sense of agency, while teachers deepened their capacity to embed culturally affirming practices into their classrooms. Although implemented in a unique context, the project offers a replicable model for schools seeking to address the academic and social–emotional needs of immigrant students through prevention oriented, identity affirming approaches. Educational systems are called to invest in and scale integrated, culturally affirming curricula that equip all students, especially those who are newly arrived with the academic and social and emotional knowledge and opportunities they need to thrive.

## Figures and Tables

**Table 1 behavsci-15-01254-t001:** Comparison of Transformative and Traditional Social–Emotional Learning.

T-SEL Feature	Connection to SEL
Identity	Focused on the development of self-awareness of one’s cultural upbringing, understanding of intersectionality and positionality
Agency	Includes elements of self-management, intentional action, and fostering hopeful action
Belonging	Tied to relationship-development and reflects a sense of connectedness to one’s environment, feeling respected and included
Collaborative Problem-Solving	Involves shared and responsible decision-making with an understanding of citizenship with current contexts
Curiosity	Helps in building social awareness and decision-making through thoughtful discourse and analysis

**Table 2 behavsci-15-01254-t002:** Core Units, Key Questions, and Mental Health Literacy Objectives.

Core Units	Key Question	Mental Health Literacy Learning Objectives
Introduction into the Units	What supports the health and wellbeing of immigrant youth?	Define immigrants and reasons for migration Describe stages and challenges of the migration journey Identify what immigrant youth need to thrive
Hope	Why Does Hope Matter?	Define hope and its key components Explain how hope supports wellbeing and resilience Apply strategies to build and sustain hope
Identity	What experiences have shaped me into the person that I am today?	Define identity and how it develops over time Identify key influences on identity Explain how identity impacts mental health and wellbeing
Social Conditions	How do our environments impact us?	Explain why health outcomes differ across individuals and groups Describe how environmental factors influence health Define social determinants of health and provide relevant examples
Agency	What supports the health and wellbeing of immigrant youth?	Define health and wellbeing across key dimensions Identify factors that influence health Explain how culture and social conditions shape health Describe ways to build individual and community health agency

**Table 3 behavsci-15-01254-t003:** Percentage of correct responses on Pre- and Post-Intervention Knowledge and Attitudes Survey for AY 2022–2023 (N = 39 Pre; N = 32 Post).

Item	Pre	Post
Knowledge Items (% Correct)		
Health includes physical, mental, and social health	95%	100%
Our health is influenced by biology, psychology, and social environment	92%	100%
Mental health influences how we think, feel, and act	95%	100%
Everyone has mental health	87%	97%
Stress is a normal reaction to challenges	77%	90%
Brain can grow/change with new experiences	90%	90%
Social conditions impact immigrant health	90%	100%
Maintaining my culture helps me stay healthy	64%	100%
Immigrants are healthier, more educated, and commit less crime	38%	75%
Being resilient helps us be healthy	74%	97%
Attitude Items (1 = Strongly Disagree; 4 = Strongly Agree)		
Importance of learning about health	M = 3.5	M = 3.8
Importance of knowing coping strategies	M = 3.3	M = 3.7
Knowledge of strategies to help self	M = 2.8	M = 3.2
Knowing where to seek help	M = 2.8	M = 3.3
Belief in making a community difference	M = 2.7	M = 3.1

## Data Availability

The original contributions presented in this study are included in the article. Further inquiries can be directed to the corresponding author.

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
