# Peer review of "Bridging Disciplines: Integrating Mental Health and Education to Promote Immigrant Student Wellbeing"

_behavsci, 2025, doi:10.3390/bs15091254_

Round 1
Reviewer 1 Report
Comments and Suggestions for Authors
This review is for the manuscript, Bridging Disciplines: Integrating Mental Health and Education to Promote Immigrant Student Wellbeing. I complete research with mental wellbeing in education; therefore, this paper was of particular interest to me.
This paper reviews a socioemotional learning program for culturally diverse students during their 12th year of school. The findings of this study are interesting and timely as an emerging area of research. Collectively, the paper is well written and interesting. However, several areas need to be addressed prior to moving forward with the manuscript.
Introduction
In-text Citations: There are several areas throughout the introduction that need citation support. For example, the statistic in line 31 lacks a reference. This occurs in lines 41, 45, 51-52, 67-38, 69-70, 78, 85-86, 98-99.
The year for the in-text citation in lines 61-63 should be placed after the agency, i.e., U.S. Department of Education (2023).
Tables: Throughout the paper, the contents of the tables are centered. This works well for the column title, but it makes reading the lines difficult. Consider left-aligning the table contents.
Headers are underused throughout the manuscript. Several areas could benefit from the addition of secondary headers: Introduction, Results, and Discussion.
Section 2.6: The units are first in italics in line 323, then in normal font in 327, and in quotes in line 333. Consider maintaining one mechanism for these units, either italics or normal font.
Table 3: Were statistical tests conducted to determine the significance between pre- and post-test results? Also, consider addressing the scores in more detail in the preceding paragraph.
Table 3: For the mean scores, what were the max scores? Consider elaborating on the surveys, the items, and how they were scored, and referencing them in the appendix.
Qualitative Data: This section needs restructuring.
More detail needs to be provided on how the statements were reviewed. At present, there is a series of quotes organized by topic. Address if more formal analysis was conducted, what coding took place, and how reliability was accounted for.
Discussion: This section also needs restructuring. At present, it is a high-level review of the study. The findings of this study need to be situated within current research, highlighting how they differ, are supported, or are in conflict with other studies.
Author Response
Reviewer 1 Comments and Responses:
Comment: In text citation for line 31.
Response: Added citation Fabina et al., 2023
Comment: In text citation for line 41.
Response: Added citation: Migration Policy Institute, 2025
Comment: In text citation for line 45.
Response: Added citation: Silva et al., 2022
Comment: In text citation for lines 51–52.
Response: Added citation: Sugarman, 2023
Comment: In text citation for lines 69–70.
Response: Added citation: Patel et al., 2023
Comment: In text citation for line 78.
Response: Added citation: Centers for Disease Control and Prevention, 2024
Comment: In text citation for lines 85–86.
Response: Added citation: Rousseau & Guzder (2008).
Comment: In text citation for lines 98–99.
Response: Added citations: Catalano et al. (2012); Elias & Hatzichristou (2016).
Comment:The year for the in-text citation in lines 61-63 should be placed after the agency, i.e., U.S. Department of Education (2023).
Response: Corrected: U.S. Department of Education (2023).
Comment: Tables: Throughout the paper, the contents of the tables are centered. This works well for the column title, but it makes reading the lines difficult. Consider left-aligning the table contents.
Response: Revised formatting so table contents are left-aligned for easier reading.
Comment:Headers are underused throughout the manuscript. Several areas could benefit from the addition of secondary headers: Introduction, Results, and Discussion.
Response: Added additional headers and sub-headers to improve organization and clarity throughout the paper.
Comment:Section 2.6: The units are first in italics in line 323, then in normal font in 327, and in quotes in line 333. Consider maintaining one mechanism for these units, either italics or normal font.
Response: Revised for consistent formatting throughout.
Comment: Table 3: Were statistical tests conducted to determine the significance between pre- and post-test results? Also, consider addressing the scores in more detail in the preceding paragraph.
Response: Clarified that no inferential statistical analyses were conducted due to small sample size and attrition (N=39 pre; N=32 post). Results are presented descriptively. Expanded description in Results regarding scoring, changes in knowledge and attitudes, and maximum values for Likert scales.
Comment:Table 3: For the mean scores, what were the max scores? Consider elaborating on the surveys, the items, and how they were scored, and referencing them in the appendix.
Response: Added detail that attitude items were measured on a 4-point Likert scale (1 = strongly disagree to 4 = strongly agree, maximum score = 4). Expanded description of survey items and included them in the Appendix for reference.
Comment: Qualitative Data: This section needs restructuring. More detail needs to be provided on how the statements were reviewed. At present, there is a series of quotes organized by topic. Address if more formal analysis was conducted, what coding took place, and how reliability was accounted for.
Response: Expanded Methods to describe structured review approach, including iterative reading of responses, identification of themes, and team-based validation of findings. Clarified that formal coding was not conducted, consistent with program evaluation design.
Comment: Discussion: This section also needs restructuring. At present, it is a high-level review of the study. The findings of this study need to be situated within current research, highlighting how they differ, are supported, or are in conflict with other studies.
Response: Revised Discussion to explicitly connect findings to existing literature, noting how they align with, extend, or differ from prior research.
Reviewer 2 Report
Comments and Suggestions for Authors
Thank you for granting me the opportunity to review this article, which aims to analyze the positive effects of Interdisciplinary Project on immigrants and refugee youth enrolled in the 12th grade English language Arts course in New England. The authors have clearly articulated the background and implementation procedures of the project; however, the research design appears insufficient to adequately support the drawn conclusions. There are several issues that the authors should meticulously address:
1. As the title of the paper suggests, the treatment strategy of this project is “integrating mental health and education”. It primarily involves collaborative interventions by school-based psychologists and English teachers aimed at supporting immigrant students. However, the authors have not clearly delineated how the intervention strategies employed by these professionals within the scope of the project differ from their routine professional responsibilities. This lack of distinction may lead to confusion between the effects attributable to the project's intervention and those stemming from regular educational practices.
2. The intervention objective of this project is to enhance the immigrant students’ well-being, a concept that encompasses a broad and multifaceted set of attributes. However, the paper lacks a detailed explanation of the operational framework of “well-being”. Based on the section “3.1 Results”, it appears that the authors have operationalized well-being using only two measurement indicators: students' mental health knowledge and students' attitudes toward the curriculum. These two indicators seem to be insufficient to fully capture the complex dimensions of well-being.
3. With regard to methodology, the statistical analysis presented in Table 3 does not include a significance test for the differences observed before and after the intervention. Consequently, the current material is insufficient to confirm a statistically significant impact of the intervention.
4. The authors should provide additional clarification regarding whether there are significant differences in well-being between the intervention group (i.e., immigrant students) and the non-immigrant student group. This would help underscore the importance and relevance of the current study.
5. In the discussion and conclusion section, the authors fail to explore the theoretical significance of this study.
In summary, although the topic of this manuscript is attractive, the current version of the paper does not fully meet the rigorous quality standards expected by this journal.
Author Response
Reviewer 2 Comments and Responses:
Comment:As the title of the paper suggests, the treatment strategy of this project is “integrating mental health and education”. It primarily involves collaborative interventions by school-based psychologists and English teachers aimed at supporting immigrant students. However, the authors have not clearly delineated how the intervention strategies employed by these professionals within the scope of the project differ from their routine professional responsibilities. This lack of distinction may lead to confusion between the effects attributable to the project's intervention and those stemming from regular educational practices.
Response: Added clarification in “Interdisciplinary Project Team” and “Implementation” sections. We now specify that the psychologist’s role expanded beyond individual therapy and consultation to include co-developing curriculum and teaching classroom lessons, while teachers moved beyond standard ELA delivery to embed mental health literacy and culturally sustaining pedagogy.
Comment: The intervention objective of this project is to enhance the immigrant students’ well-being, a concept that encompasses a broad and multifaceted set of attributes. However, the paper lacks a detailed explanation of the operational framework of “well-being”. Based on the section “3.1 Results”, it appears that the authors have operationalized well-being using only two measurement indicators: students' mental health knowledge and students' attitudes toward the curriculum. These two indicators seem to be insufficient to fully capture the complex dimensions of well-being.
Response: Revised Methods (Pre/Post Survey) to clarify that “wellbeing” was operationalized in a limited way through proximal indicators: knowledge of mental health concepts and attitudes toward help-seeking and community health.
Comment: With regard to methodology, the statistical analysis presented in Table 3 does not include a significance test for the differences observed before and after the intervention. Consequently, the current material is insufficient to confirm a statistically significant impact of the intervention.
Response: This is addressed under Reviewer 1. Clarified rationale (small sample, attrition, descriptive focus).
Comment:The authors should provide additional clarification regarding whether there are significant differences in well-being between the intervention group (i.e., immigrant students) and the non-immigrant student group. This would help underscore the importance and relevance of the current study.
Response: Clarified that the study was conducted in a school serving only immigrant and refugee students; thus, no non-immigrant comparison group was available. Our focus was on feasibility, acceptability, and curriculum development in this specific context.
Comment:In the discussion and conclusion section, the authors fail to explore the theoretical significance of this study.
Response: Restructured the Discussion and Conclusion to more clearly situate the findings within theoretical frameworks (T-SEL, SDOH, culturally sustaining pedagogy, trauma-informed practice) and highlight the broader theoretical contribution of interdisciplinary partnerships in school mental health.
Round 2
Reviewer 1 Report
Comments and Suggestions for Authors
You have done a great job incorporating the feedback and improving the quality of the paper. Good work!
Reviewer 2 Report
Comments and Suggestions for Authors
The authors have addressed the previously raised issues through targeted revisions, thereby ensuring that the current version fulfills the requirements for publication.